# Phonotaxis in Male Field Crickets: The Role of Flight Experience, Serotonin and Octopamine Neurotransmission

**DOI:** 10.3390/insects16090887

**Published:** 2025-08-25

**Authors:** Maxim Mezheritskiy, Dmitry Vorontsov, Varvara Dyakonova

**Affiliations:** Koltzov Institute of Developmental Biology of the Russian Academy of Sciences, Moscow 119334, Russia; d.vorontsov@idbras.ru (D.V.); dyakonova.varvara@idbras.ru (V.D.)

**Keywords:** phonotactic behavior, male crickets, phonotaxis, 5-HTP, serotonin, flight, insects, aggregation, octopamine, monoamines, conspecific attraction

## Abstract

Field crickets are not eusocial insects like ants or bees, but they do exhibit advanced social behavior. They use sound to find and communicate with each other. The males of *Gryllus bimaculatus* crickets produce a calling signal, to which the females respond with phonotaxis, moving towards the source of the sound. It is only natural that phonotaxis was mainly studied in female crickets. However, it is known that males also have a phonotactic response to a calling signal. This allows the male to silently move toward another singing male and to intercept the approaching females. It was widely believed that the male phonotaxis is significantly weaker than the female phonotaxis. However, the data obtained in our experiments disprove this. In our experiments, the phonotaxis of males and females did not differ in any of the measured parameters. Additionally, the mechanisms of phonotaxis modulation were similar in both sexes. Previous flight experience significantly increased phonotaxis in both males and females. Furthermore, flight-induced activation of phonotaxis appeared to be similarly associated with serotonergic neurotransmission in both sexes. Interestingly, studies on locusts and fruit flies have shown that social attraction, or attraction to conspecifics, is also based on serotonergic and, more generally, monoaminergic mechanisms. Considering phonotaxis as a kind of social attraction reveals parallels and suggests evolutionary conservatism in the serotonergic mechanisms of attraction and repulsion in insects.

## 1. Introduction

It is well known that many insects use intraspecific communication [1,2,3]. In Orthoptera, the diversity of their social and sexual behavior is largely associated with acoustic signaling [3,4,5,6]. In the field crickets (Orthoptera: Gryllidae), a male produces a calling song by rubbing his forewings while a silent female travels towards the source of the sound. This behavior is called phonotaxis [3,4]. In crickets, it has been shown that the acoustic signal alone is sufficient to activate phonotactic behavior [7]. Phonotaxis is classically considered the initial stage of mating behavior, which precedes copulation [8]. The vast majority of studies of phonotactic behavior have been performed on female crickets.

However, as has already been shown in different species of crickets, both under natural and experimental conditions, not only females but also male crickets demonstrate phonotaxis to the conspecific calling song [9,10,11,12]. The male phonotaxis is believed to serve its own biological functions. For example, it increases local aggregation of insects of both sexes in the vicinity of a singing male and thus promotes intraspecific competition between males [1,13]. Even the singing male crickets tend to stay close to each other, but at a respectful distance [14]. However, a larger cluster of singing males can also attract unwanted attention. Silent males within these clusters, being less exposed to predators and parasites, are able to intercept females during their movement towards the singing male. This so-called satellite behavior may also increase the reproduction and survival opportunities [3,15].

A few studies have shown that behavioral response of males to the conspecific calling song is significantly weaker than that of females [11,12,16]. This difference, although seeming quite reasonable, has been discovered in experimental conditions different from natural ones, namely in a trackball system where a cricket is glued to a holder and placed on a sphere that rotates when the cricket walks. Based on our observations, we suggested that male crickets may behave differently when walking on the solid ground. Testing this was the primary goal of our study.

At the same time, it is clear that the social behavior of crickets is greatly influenced by past experience, external and internal factors [17,18,19,20,21,22,23,24]. This also applies to phonotaxis and female mate choice [25,26,27,28,29]. For example, the younger the adult female and the higher the population density of males, the higher her choosiness [30]. In the presence of the predator cues or parasitic infection, the female choosiness is decreased [31,32]. Also, the intensity of phonotactic behavior depends upon the mating status [33,34]. Thus, well-understood physiological factors, or factors associated with exposure to certain external stimuli during postnatal development, are involved in the modulation of phonotactic behavior and the female mate choice. In addition, we recently demonstrated that flight, a natural behavior widespread among insects, can significantly enhance the intensity of phonotaxis in female crickets. Previous flight experience increased the intensity of phonotaxis in all measured parameters in females of *Gryllus bimaculatus* [35]. Whether flight modulates the phonotactic behavior of males remained unknown. In this study, we aimed to address this question.

Despite the fact that the two-spotted cricket, *G. bimaculatus*, is a well-known model animal in neuroethological studies [36,37,38,39,40,41,42,43,44,45], the neurotransmitter mechanisms of phonotaxis and various ways of its natural modulation are poorly understood. At the same time, it is known that changes in the monoamine balance (especially that of serotonin and octopamine) in insects often correlate with changes in their behavioral state [46,47,48,49,50,51,52,53,54]. Indeed, we have previously shown that, in females, flight resulted in an increase in serotonin concentration in the metathoracic ganglion [55]. Injection of the serotonin precursor (5-HTP) also enhanced the phonotactic response of females, while serotonin depletion abolished it. Thus, flight-induced serotonergic modulation is likely to explain the activation of positive phonotactic response. Octopaminergic modulation of phonotaxis in female crickets is less clear and is rather suppressive [55,56]. The role of monoamines in phonotactic behavior in male crickets has not been previously studied at all.

The aim of this study was to measure the phonotactic response and its modulation in freely moving male crickets (*Gryllus bimaculatus*) using a recently developed experimental paradigm [35,55]. First, we compared the phonotactic responses of males and females. Second, we tested whether flight experience modulates the phonotactic behavior of male crickets as it does in females. Third, we used pharmacological approaches to compare the monoaminergic modulation of phonotaxis in males and females.

## 2. Materials and Methods

### 2.1. Animals

Crickets (*Gryllus bimaculatus* De Geer) were reared and maintained in a laboratory colony at 25–27 °C, 65–75% relative humidity, a 12:12 light/dark cycle, and fed carrots, lettuce, dried Gammarus, wheat bran and water ad libitum. After the last molt (last instar) and two weeks before the experiment, adult males and females were taken from a colony and were kept in separate groups for two weeks in order to prevent mating [33]. One day before the experiment, all crickets were placed individually in plastic containers in order to eliminate differences associated with previous social experience [57]. During solitary confinement in separate containers, the crickets were fed fresh lettuce and carrots. Each cricket participated in a single experiment.

### 2.2. Behavioral Test Equipment

We used a spacious experimental arena (150 cm × 150 cm) to study walking phonotactic behavior (Figure 1A). The floor of the arena was made of plywood with a polyurethane matte non-slip coating. The 20 cm high walls were made of white fabric stretched between holders in each corner. The loudspeaker was located behind one of the walls, to the left of the starting position of the cricket (Figure 1B). The loudspeaker (Teac TE-T15) (Teac, Tokyo, Japan), powered by an integrated amplifier (Dynavox DA-30) (Dynavox Electronics SA, Düdingen, Switzerland) with a high-pass filter, broadcast the calling song of a male cricket, which was pre-recorded in the colony. The amplitude of the sound inside the arena was 85 dB SPL at a distance of 20 cm from the speaker, approximately at the same level as when recording sound in the colony. The arena was evenly illuminated by four adjustable LED panels (OpenScience LLC, Krasnogorsk, Russia) located 1.5 m above it, providing illumination of ca. 290–300 lx at the center. The TIS DMK 23GV024 video camera (The Imaging Source, Charlotte, NC, USA) was placed at a height of 1.5 m above the arena. The movements of the cricket were recorded at 25 frames per second using the IC Capture software (4.0) (The Imaging Source, Charlotte, NC, USA). The method was previously described in detail [35,55].

### 2.3. Behavioral Analysis

At the beginning of each experiment, the container with the cricket was manually brought to the experimental arena, opened, and carefully placed sideways against the wall of the arena into the start zone (Figure 1B), opening to the center of the arena so that the cricket could leave freely. The duration of the experiment was limited to 10 min, but it stopped earlier if the cricket, uninterested in the calling song, escaped from the arena by climbing over a fabric wall. In this way we attempted to combine a relative freedom for animals with a controlled experimental environment, allowing a cricket to decide when to enter the experiment and when to complete it. Their decisions are suggested to reflect the motivation and individual context-dependent responsiveness to the calling song of another male.

Within the arena, a virtual zone measuring 34 × 10 cm was delineated adjacent to the wall and the speaker positioned behind it, designated as the “speaker zone” (SZ, Figure 1B). The phonotactic behavior was interpreted as attraction to the SZ, measured as three different parameters: (1) the total time the cricket spent in the SZ; (2) the number of times the cricket entered the SZ (in other words, the number of approaches to the loudspeaker); (3) the distance to the SZ averaged along the track, reflecting the attitude towards the calling song. We also evaluated the average locomotion speed to check whether the changes in the above parameters may be explained by alterations of the cricket condition, in general, rather than by specific effects on phonotaxis. All parameters were extracted from the data of videotracking, performed using the EthoVision XT 13 software. Tracks were manually cleared of errors and interference.

### 2.4. Tethered Flight

A cricket was adhered to a holder by its dorsal thoracic segment and then suspended for three minutes in the fan-generated air flow to induce flight behavior, as in previous research [35,39,55,58]. In the control group, crickets were stressed by the handling procedure, which consisted of picking the cricket up by the lateral portions of the thoracic segment. Then, after manipulation, both control and experimental crickets were placed back into their individual containers for two minutes to relieve acute stress. After that, the container with the cricket was placed into the experimental arena.

### 2.5. Pharmacological Treatments

We used the octopamine receptor agonist chlordimeform (CDM) to induce octopaminergic signaling. To boost serotonin synthesis, 5-hydroxytryptophan (5-HTP), a metabolic precursor of serotonin, was used. All of these substances have been used in *G. bimaculatus* previously [38,39,59,60,61]. All substances used in this study were dissolved in an insect saline (in mmol/L: 140 NaCl, 10 KCl, 4.76 NaHCO_3_, 2 NaH_2_PO_4_·2H_2_O, 4.2 MgCl_2_, 2.7 CaCl_2_, pH 7.0). The control crickets received a saline injection accordingly. Injections were made into the hemolymph (abdominal cavity). The detailed information on dosages and procedures is presented in Table 1 and Figure 1C.

### 2.6. Statistical Analysis

Statistical analysis were performed using the PAST software (PAST: paleontological statistics software package for education and data analysis version 2.09., 2001, University of Oslo, Oslo, Norway) [62]. The nonparametric Mann–Whitney U test was used to assess statistical significance between two independent samples. The effect size was calculated using the rank-biserial correlation coefficient (r).

## 3. Results

### 3.1. Phonotactic Behavior of Intact Female and Male Crickets

We found no significant difference between males (*n* = 119) and females (*n* = 123) in any parameters of phonotactic behavior (Figure 2). Additionally, nine (from 128) males and six (from 128) females stayed inside the container for the maximal time of observation (10 min) and were excluded from the experiment. In total, 88 males and 85 females visited the SZ, while 31 males and 38 females escaped the arena without visiting the SZ.

Males and females that approached the source of the calling signal, remained near it for a similar amount of time (*p* = 0.7; Z = 0.27; Mann–Whitney U-test, Figure 2C). Another strategy that we observed was repeated leaving the SZ and returning to it. We assume that the number of times the animal repeatedly returned to the speaker is also important for assessing the strength of phonotaxis. The number of times the crickets repeatedly entered the SZ was not different between males and females (*p* = 0.8; Z = 0.24; Mann–Whitney U-test, Figure 2D). The average distance to the SZ measured along the whole track was also similar (*p* = 0.3; Z = 0.85; Mann–Whitney U-test, Figure 2E). Female crickets had a higher average speed of locomotion than males, but this difference was not statistically significant (*p* = 0.1; Z = 1.36; Mann–Whitney U-test).

### 3.2. Effects of Flight Experience on the Phonotactic Behavior of Male Crickets

Following the flight, male crickets spent significantly more time in the SZ (*n* = 10, *n* = 10; *p* = 0.01; Z = 2.46; r = 0.55; Mann–Whitney U-test, Figure 3C) compared to non-flyers. The number of visits to the SZ was higher in flyers (*p* = 0.02; Z = 0.39; r = 0.08; Mann–Whitney U-test, Figure 3D). They also kept lower average distance to the SZ (*p* = 0.003; Z = 2.75; r = 0.61; Mann–Whitney U-test, Figure 3E). At the same time, the average locomotion speed was lower in flyers (*p* = 0.01; Z = 2.45; r = 0.54; Mann–Whitney U-test).

### 3.3. Effect of Serotonin Precursor (5-HTP, 20 mM) on Phonotactic Behavior of Male Crickets

After the 5-HTP injection the crickets spent significantly more time in the SZ (*n* = 10, *n* = 10; *p* = 0.004; Z = 2.81; r = 0.62; Mann–Whitney U-test, Figure 4C), visited it more often (*p* = 0.007; Z = 2.60; r = 0.58; Mann–Whitney U-test, Figure 4D) and kept lower average distance to the SZ (*p* = 0.02; Z = 2.15; r = 0.48; Mann–Whitney U-test, Figure 4E). The locomotion speed was lower in the 5-HTP-injected crickets (*p* = 0.05; Z = 1.92; r = 0.42; Mann–Whitney U-test).

### 3.4. Effects of the Octopamine Receptor Agonist Chlordimeform on the Phonotaxis of Male Crickets

#### 3.4.1. Chlordimeform 2 mM

Crickets injected with either CDM 2 mM or saline demonstrated no significant differences in time they spent in the SZ (*n* = 17, *n* = 16; *p* = 0.09; Z = 1.66; Mann–Whitney U-test, Figure 5C), in the number of visits to the SZ (*p* = 0.3; Z = 0.99; Mann–Whitney U-test, Figure 5D) and in the average distance to the SZ (*p* = 0.08; Z = 1.74; Mann–Whitney U-test, Figure 5E). The locomotion speed was lower in the CDM-injected crickets (*p* = 0.03; Z = 2.14; r = 0.37; Mann–Whitney U-test). Thus, administration of a low dose of CDM only affected the locomotion speed.

#### 3.4.2. Chlordimeform 4 mM

The injection of higher (4 mM) dose of CDM resulted in reduction in time spent in the SZ (*n* = 17, *n* = 15; *p* = 0.0004; Z = 3.29; r = 0.58; Mann–Whitney U-test, Figure 6C) and of the number of visits to the SZ (*p* = 0.0004; Z = 3.33; r = 0.58; Mann–Whitney U-test, Figure 6D). The average distance to the SZ did not differ between the two groups (*p* = 0.5; Z = 0.56; Mann–Whitney U-test, Figure 6E). The average speed of locomotion was lower in the CDM-injected crickets (*p* = 0.005; Z = 2.75; r = 0.48; Mann–Whitney U-test).

## 4. Discussion

### 4.1. Phonotaxis of Male Crickets

Competition or fighting for food or space, and courtship behavior are typical for *G. bimaculatus* [8,63]. Crickets are even capable of social learning [64]. However, animals must first meet each other to exhibit social behavior. The number of interactions increases if the calling signal affects both sexes. Females may prefer male aggregations because this increases their mating choice and may indicate a nearby resource availability [14].

We confirmed the presence of male phonotaxis in *Gryllus bimaculatus* by conducting a study with a relatively large number of crickets. Under our experimental conditions, there were no statistically significant differences in the phonotactic responses of male and female crickets for any of the measured parameters. Both males and females were attracted to the speaker that broadcasted the male calling song. They spent similar amount of time near the speaker. Males also approached the speaker as often as females did. While moving around the arena, they maintained the same average distance from the speaker. At the same time, no difference in locomotion speed was found between males and females.

In this regard, our findings contradict earlier studies by other researchers that demonstrated a more pronounced phonotaxis in female crickets [11,12,16]. One of the aforementioned studies [11] was conducted on a different species, *Gryllus texensis*, which may be the primary reason for the discrepancy in results and conclusions. However, other studies on *G. bimaculatus* reported that only 20% [16] or 25% [12] of males responded to the male calling signal by phonotaxis. This is in dramatic contrast to the results of our study, in which 74% of males exhibited robust phonotactic behavior. Below we suggest a number of factors that may be responsible for the observed discrepancy.

Firstly, crickets were obtained from different geographical populations, and their cultivation in laboratories was subject to varying conditions.

Secondly, we tried to minimize the amount of manipulation and stress. Specifically, no handling was applied to the cricket after initial isolation. The cricket exited the container and entered the arena voluntarily, not as a result of coercion or force. Upon entering the arena, the cricket was presented with a decision: to opt out of the experiment if it was not interested in the calling signal, or to proceed towards the loudspeaker. The crickets that did not exit from their home containers were subsequently excluded from the experiment. It appears that the crickets that remained in their containers were either too frightened or not healthy enough to exhibit phonotactic behavior. In this regard, home containers can be considered an artificial analog of burrows, wherein the animal, depending on its functional state, determines whether to respond to the calling signal.

Thirdly, locomotion on a solid surface, as opposed to the more classical approach to measuring of the phonotaxis by utilizing a trackball system [12,65], may serve as an additional explanation for the more pronounced phonotaxis observed in our experiments in males.

### 4.2. Flight and Male Phonotaxis

Regardless of the behavioral strategy that a male uses to attract a female (e.g., a calling song or a satellite behavior), it is clear that both males and females are interested in effective phonotactic mechanisms. Flight has been shown to enhance phonotaxis in both female [35,55] and male crickets (present study). After flying, males were significantly more attracted to the source of the calling song than in the control group. They spent more time near the source and returned to it repeatedly. Thus, flight positively affects phonotaxis in crickets of both sexes. However, the effects of flight on the subsequent reproductive activity of *G. bimaculatus* females, such as the mate choice, the number of matings, the number and quality of eggs, and offspring survival are currently unknown. More is known about the effects of flight on male crickets [39,57]. After flight, the males of *G. bimaculatus* behave more aggressively toward other males, which is associated with competition for space. They also court females more actively [58]. Flight experience leads to an increase in the number and duration of courtship and calling songs produced by a male, thus promoting reproductive success [58].

It could be hypothesized that, in the wild, crickets that fly and subsequently demonstrate the enhanced phonotaxis and courtship behavior can be more successful in reproduction. Thus, flight behavior could become fixed in subsequent generations. This could explain the deterioration of flight ability in the laboratory crickets of both sexes (reported by Lorenz [66], and in agreement with our own observations). In a laboratory colony, where crickets live in close quarters and have no opportunity to fly, flight ceases to influence reproductive success. Thus, the likelihood of “good flyers” emerging in subsequent generations is reduced.

There is a complex relationship between flight and reproductive functions. The ability to fly, migration or dispersal are associated with either the deterioration in insect reproductive functions (the so-called oogenesis-flight syndrome) or improvement in reproductive fitness depending on the species, sex, stage of development, etc. [66,67,68,69]. In some species of crickets, the flight experience and the ability to fly are inversely correlated to reproductive functions [66,70,71,72,73,74].

At the same time, it was shown that, in long-winged female crickets *Velarifictorus asperses* and in *G. texensis*, flight promoted reproductive development [69,75,76]. In *V. asperses*, flight increases the weight of accessory glands, prolongs the singing time and even changes the song structure, which increases the attraction of females [76,77]. The acoustic behavior of flight-capable males of *G. texensis* is more pronounced than that of flightless morphs [78]. Also, the long-winged male morphs of *G. texensis* exhibit an increased courtship behavior after flight [75]. Thus, the behavioral and physiological states after flight may, in some cases, compensate for the deterioration of reproductive functions in flying individuals. It may also reflect a shift from migration and/or dispersal to reproduction [69,76,77]. Our current data are consistent with this hypothesis.

### 4.3. Serotonin, Octopamine and Phonotaxis

We recently demonstrated that natural enhancement of phonotaxis in females of *G. bimaculatus* is associated with serotonergic signaling [55]. Here we show that the injection of the serotonin metabolic precursor, 5-HTP, enhances phonotaxis in male crickets also. Male crickets that received 5-HTP spent significantly more time in the arena zone near the speaker (speaker zone, SZ), entered the SZ and kept less distance to the speaker on average, compared to the saline-injected control group. We suggest that the increase in serotonin concentration in certain parts of the nervous system of both female and male crickets plays a key role in enhancing phonotaxis.

Serotonin plays an important role in social interactions in crickets [38,43,59,60,61,79,80]. Pharmacological depletion of serotonin increases the escape reaction and also reduces the ability to gain dominant status [38]. Interestingly, the level of serotonin in the brain of males decreases naturally during the fight with another male, but the winner raises the level of serotonin in his brain by producing an aggressive acoustic signal, or «rival song». In contrast, the loser’s serotonin level remains reduced [43,79]. The question of behavioral meaning of this mechanism remains open [43,79]. Why does a winner need to increase serotonin? In his report Akustische Verständigung im Tierreich, the German neuroethologist Franz Huber (1925–2017) mentioned that phonotaxis is suppressed in male crickets after they lose a fight [81]. We assume that the suppression of phonotaxis is probably associated with a decrease in serotonin concentration in the central nervous system. An increase in serotonin level (or retention of its level compared to the loser) in the winner cricket may positively affect subsequent phonotaxis, aggression, and courtship behavior [59,60].

In some cases, serotonin and octopamine oppositely regulate behavior and physiological functions in insects [82]. Our results suggest that this is also true for phonotaxis in *G. bimaculatus*. The octopamine receptor agonist chlordimeform (CDM) has no effect at a concentration of 2 mM and significantly reduces phonotactic behavior at a concentration of 4 mM in all measured parameters. We did not observe any toxic effects: the crickets appeared healthy and were able to exit their containers into the experimental arena and to move around freely. However, they did not seek out the source of the calling signal. Of course, it is difficult to draw a reliable conclusion about the role of octopamine in phonotaxis based on the effects of CDM alone. However, the present data are supported by other studies. Previously, we demonstrated that CDM significantly reduced phonotaxis in females and that the octopamine receptor antagonist epinastine significantly improved phonotaxis in female crickets [55,83]. Injection of octopamine into the thoracic ganglion of female house crickets *Acheta domesticus* also impaired phonotaxis [56,84]. Thus, our data support the idea that behavioral modulation of phonotaxis by flight is based on monoaminergic mechanisms. The enhancement of phonotaxis is associated with increased activity of serotonergic system. The role of octopamine is less clear but appears to be inhibitory. Interestingly, we found no significant differences in the monoaminergic modulation of phonotaxis between males and females, which strengthens the conclusion that the phonotactic behavior is similar in both sexes of *G. bimaculatus*. At the same time, the “flight”, “serotonin” and “CDM” groups demonstrated the decrease in the average locomotion speed. Thus, there is no simple, obvious correlation between the average locomotion speed and the strength of phonotaxis.

### 4.4. Monoamines and Aggregation

The behavioral and physiological mechanisms underlying aggregation in insects may vary depending on the species. It is noteworthy that serotonergic modulation of aggregation-related behavior is not exclusive to the cricket *G. bimaculatus.* It is also observed in the closely related species, the locusts *Schistocerca gregaria* and *Locusta migratoria*, as well as in the taxonomically distant fruit fly *Drosophila melanogaster* [50]. It was shown that the activation of serotonergic system is necessary and sufficient for social attraction (or conspecific attraction) in both *S. gregaria* and *Drosophila* [47,85,86,87,88]. In the locust *L. migratoria*, serotonin has been demonstrated to be associated with the initiation of the opposite process—the transition to a solitary form. This transition is consequently associated with a decrease in social attraction, or social avoidance [89]. In contrast, the transition to a gregarious form in *L. migratoria* is associated with dopamine, while in *S. gregaria* dopamine mediates the transition to a solitary form [90,91,92,93]. The role of octopamine in the phase transition in locusts is not easily interpreted [85,94]. The concentration of octopamine in the nervous system of *S. gregaria* exhibits significant fluctuations in accordance with the duration of a solitary individual’s presence within a crowd of conspecifics [85,95]. In *L. migratoria*, octopaminergic signaling has been demonstrated to be directly associated with the attraction behavior in response to odors, such as volatiles emanating from the bodies and feces of conspecifics. Octopamine’s metabolic precursor tyramine has been demonstrated to be associated with repulsion [96]. It is important to acknowledge that the studies of *S. gregaria* and *L. migratoria* featured notable methodological distinctions. For instance, in *S. gregaria,* serotonin and its antagonists were injected into the thoracic ganglia, whereas in *L. migratoria*, these substances were injected into the head [47,89].

However, the collective evidence from these studies highlights the significance of biogenic amine signaling in the mechanisms of attraction-repulsion behavior in insects.

### 4.5. Conclusions

This study reveals several insights into cricket phonotaxis behavior and its monoaminergic regulation:No obvious sex differences were observed neither in intensity of phonotactic responses to calling signal, nor in the modulation of this response by previous behavior or by monoamines.Phonotaxis in male crickets is enhanced by flight. While factors reducing phonotaxis in male crickets (e.g., age, illness, recent copulation) are known, enhancers were previously studied only in female crickets. Here we show that the prior flight experience boosts phonotaxis in male crickets., This may promote aggregation and may compensate for flight-induced reproductive decline (e.g., the oogenesis-flight syndrome).Serotonergic signaling is associated with enhanced phonotaxis while octopaminergic system suppresses phonotaxis both in male and female crickets.

Our research suggests many interesting avenues for further investigation. For example, it is unclear (1) whether the serotonin levels track the behavioral state, being different in call-receptive and non-receptive crickets; (2) whether flight affects the phonotaxis in other cricket species and in their grasshopper relatives; (3) whether the modulation of aggregation-related behavior by serotonin is inherent in other Gryllidae and grasshoppers, including poorly studied locust species. Through combined efforts, we can better understand some common behavioral and evolutionarily conserved neurochemical mechanisms related to attraction and repulsion in insects.

## Figures and Tables

**Figure 1 insects-16-00887-f001:**
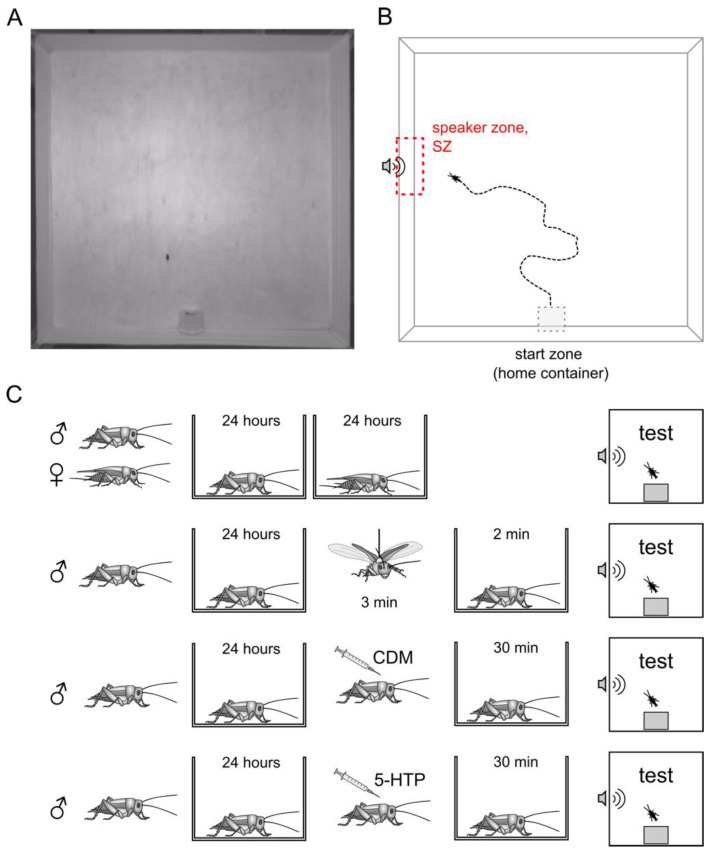
Experimental protocol. (**A**) Screenshot of a recorded video showing the experimental arena with a cricket. (**B**) Schematic picture, showing a typical track of the cricket moving from the container (start zone) towards the source of the calling song (speaker zone). (**C**) Schematic protocol of behavioral and pharmacological experiments showing the sequence of procedures, from the left to the right.

**Figure 2 insects-16-00887-f002:**
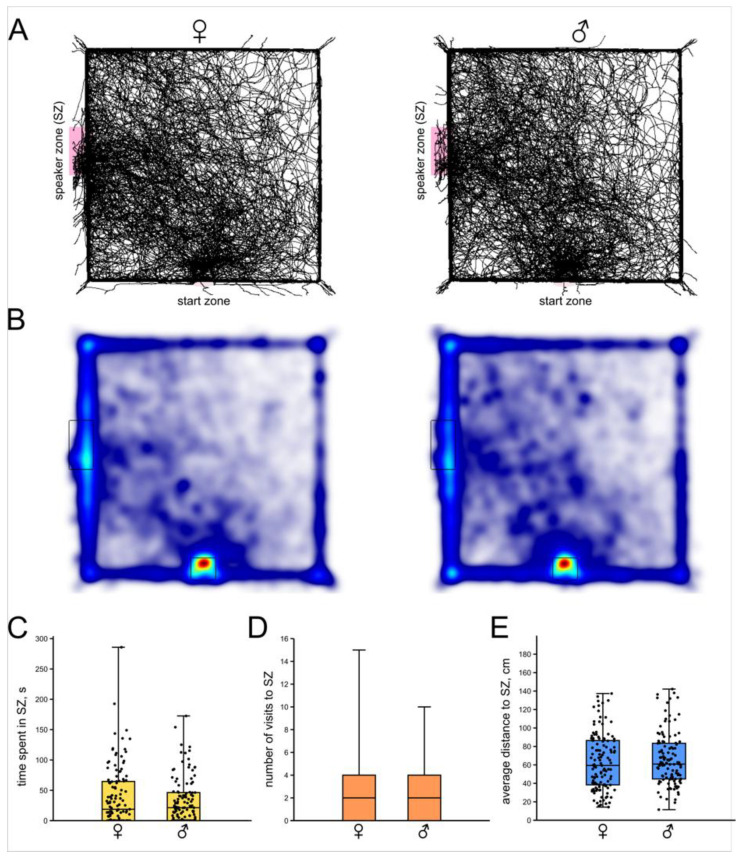
Phonotactic behavior of intact female and male crickets. (**A**) Superimposed individual tracks of female (left) and male (right) crickets. (**B**) The same, shown as heatmaps, which better represent the integral time spent in different parts of the arena, color indicates relative integral time, from blue (minimum) to red (maximum). (**C**) Time spent in the area near the speaker (speaker zone). (**D**) Number of visits to the speaker zone. (**E**) Average distance to the speaker zone. (**C**–**E**) There are no statistically significant differences between the male and female groups for any of the parameters.

**Figure 3 insects-16-00887-f003:**
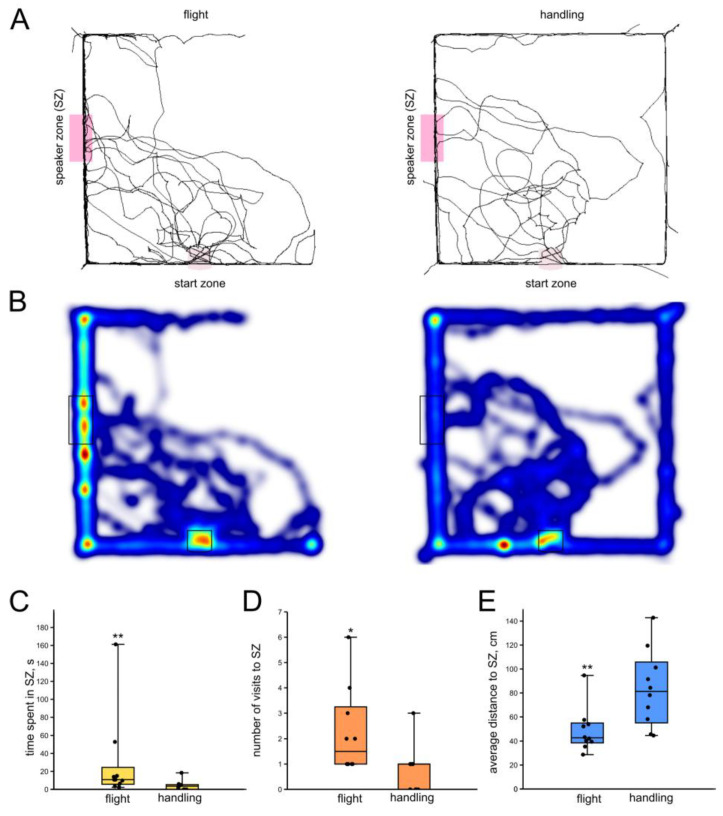
Effects of flight on the phonotactic response of male crickets. (**A**) Superimposed individual tracks of crickets after flight and in the control (handling) groups. (**B**) The same, shown as heatmaps, color indicates relative integral time, from blue (minimum) to red (maximum). (**C**) Flight increased the time spent in the speaker zone. (**D**) Flight increased the number of visits to the speaker zone. (**E**) The average distance to the speaker was lower in the post-flight group. Asterisks indicate the level of statistical significance according to the Mann–Whitney U test: * *p* ≤ 0.05, ** *p* ≤ 0.01.

**Figure 4 insects-16-00887-f004:**
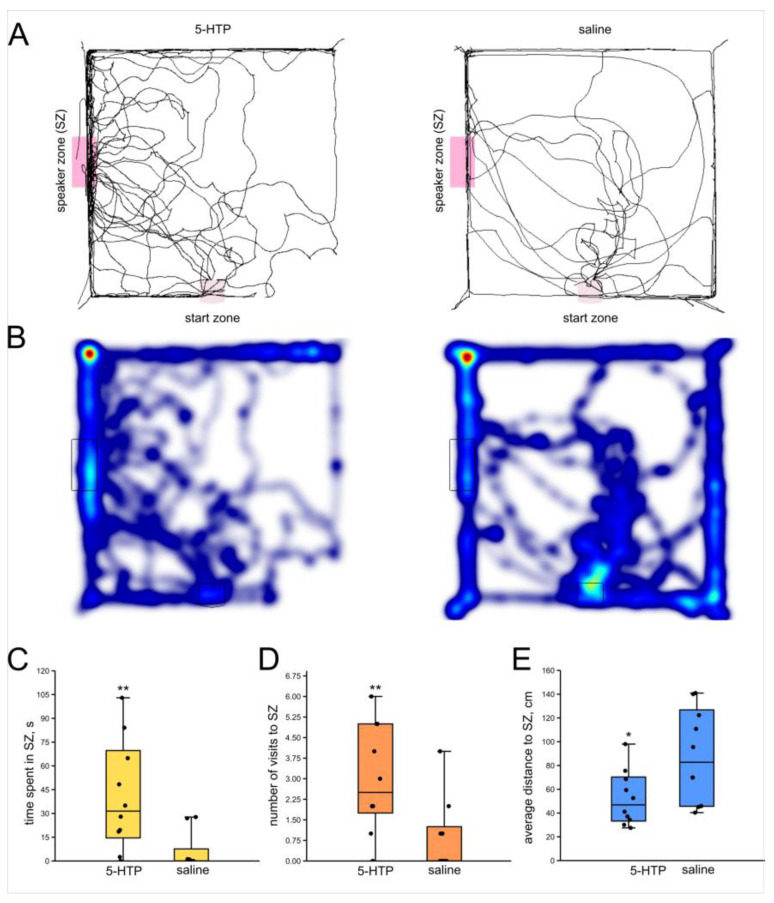
Effects of 5-HTP on the phonotactic response of male crickets. (**A**) Superimposed individual tracks of crickets after 5-HTP injection and in saline-injected control group. (**B**) The same, shown as heatmaps, color indicates relative integral time, from blue (minimum) to red (maximum). (**C**) 5-HTP increased the time spent in the speaker zone. (**D**) 5-HTP increased the number of visits to the speaker zone. (**E**) The average distance to the speaker was lower in the 5-HTP group. Asterisks indicate the level of statistical significance according to the Mann–Whitney U test: * *p* ≤ 0.05, ** *p* ≤ 0.01.

**Figure 5 insects-16-00887-f005:**
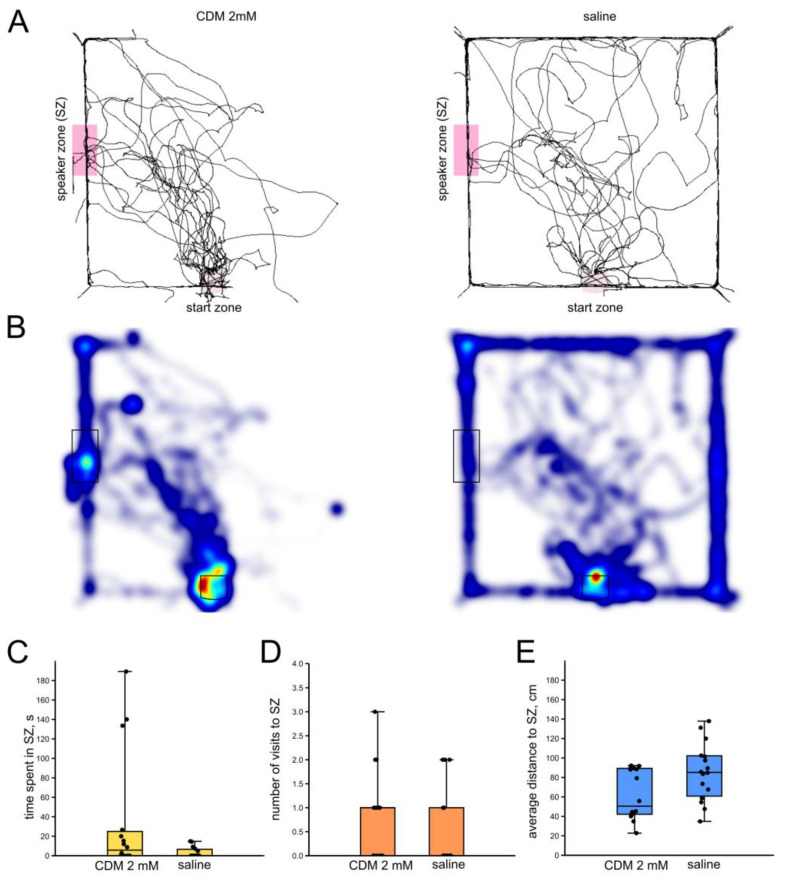
Effects of CDM (2 mM) on the phonotactic response of male crickets. (**A**) Superimposed individual tracks of crickets after CDM injection and in the saline-injected control group. (**B**) The same, shown as heatmaps, color indicates relative integral time, from blue (minimum) to red (maximum). (**C**–**E**) There are no statistically significant differences between the CDM and control groups for any of the parameters.

**Figure 6 insects-16-00887-f006:**
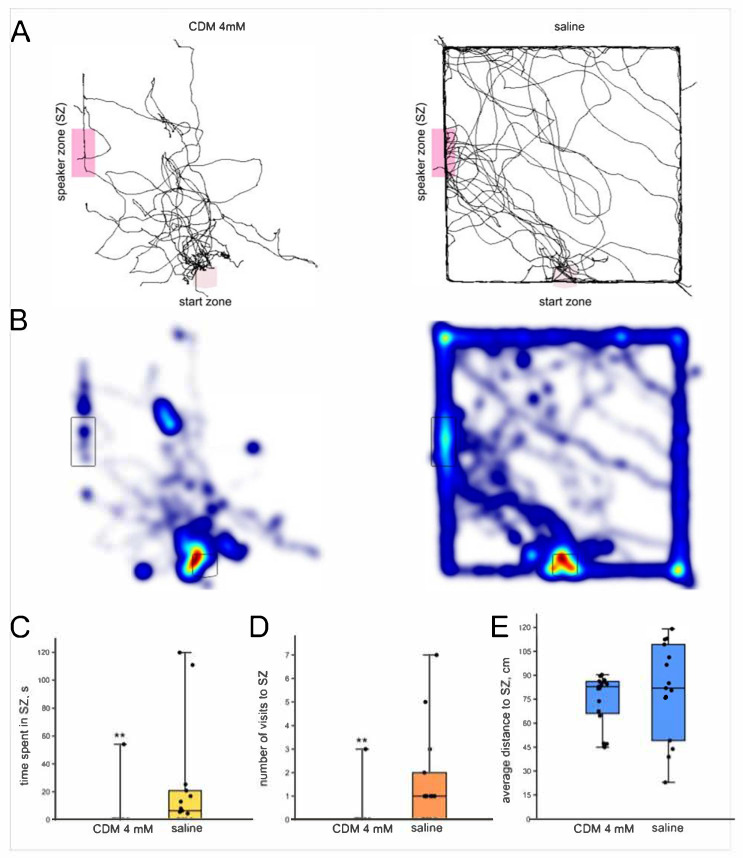
Effects of CDM (4 mM) on the phonotactic response of male crickets. (**A**) Superimposed individual tracks of crickets after CDM injection and in the saline-injected control group. (**B**) The same, shown as heatmaps, color indicates relative integral time, from blue (minimum) to red (maximum). (**C**) CDM decreased the time spent in the speaker zone. (**D**) CDM decreased the number of visits to the speaker zone. (**E**) The average distance to the speaker did not differ between groups. Asterisks indicate the level of statistical significance according to the Mann–Whitney U test: ** *p* ≤ 0.01.

**Table 1 insects-16-00887-t001:** Pharmacological treatment of crickets. See also Figure 1C.

Agent	Concentration	Volume of Injection	Vehicle	Remarks
Chlordimeform (CDM, the octopamine receptors agonist) (Sigma-Aldrich, Burlington, MA, USA)	2 mM 4 mM	100 μL	saline	Crickets were injected 30 min prior to testing. Chlordimeform acted for at least 1.5–2 h after injection.
5-hydroxytryptophan (5-HTP), immediate metabolic precursor of serotonin	20 mM	100 μL	saline	Crickets were injected 30 min prior to testing. 5-HTP acted for at least 1.5–2 h after injection.

## Data Availability

Data are contained within the article.

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
