# Peer review of "Phonotaxis in Male Field Crickets: The Role of Flight Experience, Serotonin and Octopamine Neurotransmission"

_insects, 2025, doi:10.3390/insects16090887_

Round 1

Reviewer 1 Report

Comments and Suggestions for Authors

The authors build upon their previous publication (Mezheritskiy et al., 2024) and now show that, in this experimental paradigm, male crickets exhibit comparable attraction to mating calls, with similar effects of behavioral state (flight) on this auditory behavior. After establishing their paradigm, they also perform pharmacological perturbations targeting serotonergic and octopaminergic pathways, extending their earlier work. These manipulations produce parallel results. Overall, the data presentation and analysis are solid.

One minor concern is whether the underlying behavioral state that induced potentiation is specifically flight, or whether it could be any other form of exercise or locomotion. At this stage, it is not possible to attribute the effect solely to flight (I apologize if this has already been addressed in the literature).

In their previous paper, the authors used HPLC to show an increase in serotonin levels. Including similar measurements here could strengthen the causal link. While this would be a challenging addition, HPLC might be even more informative if performed on call-receptive and non-receptive males. If serotonin levels track the behavioral trend, such a result would be invaluable.

The most intriguing aspect of this work is the discrepancy with previous literature, which reported that male crickets respond significantly less to male chirps. Here, the authors directly contradict those findings. The paper’s impact would increase if the underlying cause could be identified.

While none of the following suggestions are essential for this paper:

  • Experimental substrate has been suggested as a source of such discrepancies. Would it be worth testing different substrates (Sarmiento-Ponce et al.)?
  • The authors suggest that use of a trackball might explain the difference in male behavior. As tethering can alter animal behavior, it would be useful to support this argument with literature comparing free-moving and tethered insect behavior.

The authors have developed a strong experimental setup. I look forward to future work, especially in terms of male interception as competitive behavior. Can they reproduce this in their assays within a more social context?  Also, do really silent males tend to intercept more?

Reviewer 2 Report

Comments and Suggestions for Authors

Opinion
The text is a comprehensive, systematically structured scientific manuscript that investigates the phonotactic behavior of male Gryllus bimaculatus crickets and its monoaminergic regulation. The paper is noteworthy in several respects, although some critical remarks can also be made.

Strengths
a) Thorough presentation of scientific background: The introduction is exceptionally detailed, utilizing an extensive range of literature references that establish the relevance of the research. It presents the role of phonotaxis in both females and males, then highlights contradictions in the literature, thereby formulating a clear research question.
b) Innovative research question: Research into male cricket phonotaxis is rare, especially involving freely moving animals rather than using the classic "tethered trackball" method. This methodological approach enhances ecological validity and opens new perspectives in behavioral research.
c) Close hypothesis-experiment relationship: The research goals—male-female comparison, effect of flight, monoaminergic regulation—are clearly reflected in the results section.
d) Comprehensive discussion: The discussion thoroughly contextualizes the results, comparing them to other species and placing them within broader evolutionary and behavioral physiological frameworks. It spans a wide range from social learning to migration-reproduction trade-offs.

Weaknesses and Problematic Aspects
a) Overly long introduction and discussion: The introduction (and partially the discussion) is overloaded with information. While the thoroughness is commendable, the focus occasionally drifts, and the reader may get lost in the cited examples. Editorial suggestion: a more focused introduction and clearer segmentation.
b) Redundant repetitions: Certain information appears multiple times (e.g., flight enhances phonotaxis in females; role of serotonin). These reduce reading efficiency. Sections 4.1 and 4.3, for example, are partially parallel.
c) Statistical weaknesses: The sample size is small in some cases (e.g., 10 vs. 10), which calls into question the robustness of the results. Moreover, there is no information about statistical effect sizes or corrections for multiple testing (e.g., Bonferroni). This weakens the reliability of the results.

Stylistic and Editorial Comments
a) Language: The language is professional but at times overwrought and cumbersome, particularly due to the length of some sentences. Shortening sentences and simplifying syntax could improve readability.
b) Clear highlighting of research goals: Although the research goals are present, they are somewhat hidden in the middle of the text. It would be advisable to highlight them separately, preferably as a list at the end of the introduction, not just before the methodology section.

Scientific Significance and Future Directions
a) Novelty: This is one of the first studies to demonstrate flight-induced enhancement of phonotaxis in male crickets and its serotonergic basis. This is important not only at the species level but also allows for generalizable conclusions in insect behavioral neurobiology.
b) Applicability: Understanding the mechanisms of monoaminergic modulation may contribute, for example, to insect population control, development of model systems for artificial intelligence or robotics (phonotaxis-based navigation), and understanding evolutionary psychobiological questions (e.g., neurochemical bases of social attraction).

Summary:
The manuscript is of high scientific quality and thoroughly written, though occasionally overwritten. It could benefit from improvements in structure, terminological consistency, and statistical support. Despite its flaws, it is worthy of publication, especially if the figures and methodological details are complete and clear.
